# The Utility of Data Collected as Part of Australia’s Aboriginal and Torres Strait Islander Health Performance Framework

**DOI:** 10.3390/ijerph21030340

**Published:** 2024-03-13

**Authors:** Boyd Potts, Christopher M. Doran, Stephen Begg

**Affiliations:** 1Cluster for Resilience and Wellbeing, Appleton and Manna Institutes, Central Queensland University, Brisbane, QLD 4701, Australia; b.potts@cqu.edu.au; 2Violet Vines Marshman Centre for Rural Health Research, La Trobe University, Bendigo, VIC 3550, Australia; s.begg@latrobe.edu.au

**Keywords:** Aboriginal and Torres Strait Islander, health performance, data

## Abstract

Since 2006, the Australian Aboriginal and Torres Strait Islander Health Performance Framework (HPF) reports have provided information about Indigenous Australians’ health outcomes. The HPF was designed, in consultation with Indigenous stakeholder groups, to promote accountability and inform policy and research. This paper explores bridging the HPF as a theoretical construct and the publicly available data provided against its measures. A whole-of-framework, whole-of-system monitoring perspective was taken to summarise 289 eligible indicators at the state/territory level, organised by the HPF’s tier and group hierarchy. Data accompanying the 2017 and 2020 reports were used to compute improvement over time. Unit change and confidence indicators were developed to create an abstract but interpretable improvement score suitable for aggregation and visualisation at scale. The result is an exploratory methodology that summarises changes over time. An example dashboard visualisation is presented. The use of secondary data inevitably invites acknowledgments of what analysis cannot say, owing to methods of collection, sampling bias, or unobserved variables and the standard mantra regarding correlation not being causation (though no attempt has been made here to infer relationships between indicators, groups, or tiers). The analysis presented questions the utility of the HPF to inform healthcare reform.

## 1. Introduction

Aboriginal and Torres Strait Islander people (hereafter referred to as Indigenous Australians) account for 3.3% of the Australian population [1]. Although health expenditure for Indigenous Australians is higher than that for non-Indigenous Australians, their burden of ill health is considerably larger, with [2,3] inequalities in health outcomes arising from a number of factors including the ongoing impacts of colonisation and entrenched social and economic factors [4]. The 2008 ‘Close the Gap’ strategy initiated by the Australian Government aims to achieve equality for Indigenous Australians within a generation [5]. There are 17 National socio-economic targets across areas that have impacts on life outcomes for Indigenous Australians in health and wellbeing, education, employment, justice, safety, housing, land and waters, languages, and digital inclusion. Five of these targets relate to health and wellbeing, with a key priority being to close the health and life-expectancy gap between Indigenous and non-Indigenous Australians within a generation [5].

Since 2006, the Australian Institute of Health and Welfare (AIHW) has generated Aboriginal and Torres Strait Islander Health Performance Framework (HPF) reports that provide a range of information pertaining to Indigenous Australians’ health and wellbeing outcomes and factors influencing the performance of the health system [6]. Co-designed with Indigenous Australian stakeholder groups, the purpose of the HPF was to promote accountability of service delivery, inform Indigenous health policy and research, and encourage informed debate about Indigenous Australians’ health and health policy decision-making.

The HPF is made up of 68 measures that are hierarchically organised into groups within three tiers comprising health status and outcomes (Tier 1), determinants of health (Tier 2), and health system performance (Tier 3) (see Table 1 below for an overview of the HPF performance measures for each Tier). The AIHW is charged with the tabling of all potentially relevant and available data for each measure in the HPF, which are now available on the dedicated website https://www.indigenoushpf.gov.au/ (accessed on 30 June 2022) in the form of summary reports and accompanying data tables. This information illustrates changes that have occurred for the measures and draws implications for further improvement. It also explores differences within the Indigenous population by age, geography, and other characteristics. This is presumably intended to help identify what is working well and how to better target policy and services to meet the needs of Indigenous Australians. Indeed, information from HPF reporting has been used, at least in part, to monitor progress towards achieving the Australian Government’s Closing the Gap health targets and the Implementation Plan goals for the Aboriginal and Torres Strait Islander Health Plan 2013–2023 [5,7].

With respect to monitoring specific indicators of interest independently, the HPF provides a valuable resource. However, the questions of interest to health service providers and decision-makers are rarely, if ever, limited to measurements of individual indicators in isolation; they are more typically concerned with relationships between indicators and interventions, previous policy decisions, or other factors. For these purposes, the comprehensiveness of the HPF quickly becomes overwhelming. The 2020 HPF report, for example, is accompanied by 1401 separate data tables from over 60 different data sources. For each measure, data are provided for multiple potentially relevant indicators/variables (herein, the term “indicator” is used to distinguish between the HPF measures as theoretical constructs and the actual variables provided in the data tables). For example, under Measure 1.01—Low birth weight, data tables are provided for all births, singleton births, the Indigenous statuses of the mother and baby, birth outcomes, maternal characteristics, baby characteristics, jurisdiction, remoteness, and international comparison. Data provided for this measure are sourced from five different repositories. Across the 2020 report data, between 2 and 66 tables are provided for each measure. In some instances, the same data table is published as being relevant to more than one measure.

HPF data are published separately at national, state/territory, and remoteness levels of aggregation where available. With respect to analysing systematic effects, this limits the number of data points (observations) for any measure to a maximum of eight. Depending on missing data and aggregation of smaller jurisdictions, as few as five observations may be available for some measures. All indicators are published as single point estimates rounded to one decimal place, obfuscating differences within the rounding threshold. For example, any true values between 0.05 and 0.14 are reported as 0.1. Measures of spread (e.g., standard deviation) are not provided for most indicators. Missing or suppressed (not published—“n.p.”) values feature in many data tables. For some measures, comparisons are flagged for statistical significance or as warnings to interpret changes with caution owing to high standard errors. Each table usually contains comprehensive footnotes regarding important information on how data were collected and/or tabulated, limitations, and known biases or gaps.

Clearly, those publishing the data have sought to provide as much information as possible, but the onus is on the end user to identify which tables are useful for their purposes. Indeed, literature featuring citations for the HPF either focuses on isolated statistics of interest [8,9,10,11,12,13] or broad acknowledgement of the framework’s existence or reported themes [14,15,16,17,18]. An empirical treatment of the HPF as a connected or unified framework is not present in the current literature. For the potential of the HPF to be fully realised, a method of synthesising data from multiple sources that use different levels of measurement in a single, high-level analytical framework is required. This would extend the utility of the HPF data beyond an expansive directory of individual indicators into a whole-of-system, whole-of-framework monitoring capability. Such a method would ideally produce results that are easily communicated and understood.

The purpose of this research is to examine the usefulness of the HPF data to track improvements in Aboriginal and Torres Strait Islander health and wellbeing, determinants of health, and system performance in accordance with the framework’s tier and group structure. The manner in which HPF data are provided does not readily lend itself to robust statistical tests, and substantial additional work would be required on an indicator-by-indicator basis to appraise the quality of data and whether a suitable statistical test exists. In the absence of this clarity, a heuristic approach based on the available data was devised to analyse improvement driven by two questions: (1) Did the indicator change over time; and (2) What is the strength of evidence for any observed change? The result is an exploratory methodology that summarises changes over time. An example dashboard visualisation is presented.

## 2. Methods

### 2.1. Data

Associated data for the 2017 and 2020 HPF reports were downloaded from indigenoushpf.gov.au as Microsoft Excel workbooks. Data manipulation and computation were performed using the Python programming language, version 3.8.8. For each HPF measure, indicators that reflected two distinct time periods were selected, with this being the most common format across measures. Any indicator published at the state/territory level was eligible for inclusion.

One-hundred and sixty-three data tables from each report (326 in total) were manually transcribed (using Microsoft excel) from the downloadable workbooks to a standardised dataset format that was more suitable for analysis. Two-hundred and eighty-nine indicators were selected based on their availability in both reports at the state/territory level.

For each indicator, a decision was made subjectively on the direction of effect that indicated improvement. For rates of hospitalisation, mortality, smoking, alcohol and substance use, and service waiting times, a quantitative decrease was taken as direction of improvement. Increases in health assessments and screening, fruit/vegetable intake, physical activity, breastfeeding, community mental-health contacts and cultural identification were taken as the direction of improvement. Direction of effect assumptions for all indicators are detailed in the Appendix A.

Quantitative changes over time were calculated as the difference between published values in the 2017 and 2020 data tables for each indicator. For ordinal categorical indicators such as smoking status, a weighted sum of the rates in each category (daily smoker = 1, other smoker = 2, ex-smoker = 3, never smoked = 4) was used to examine changes over time. These were positively geared such that higher scores reflected improvement (less smoking). Where variables contained only two categories, the subjectively favourable category was selected for analysis. For example, prostate screening rates for both tested and not tested retained only the tested category. For indicators with nested categories, the broader category was retained. For example, breast screening data are provided for categories of ‘regular mammograms’ or ‘any mammogram’, with the former being a subset of the latter.

Data points published as ‘n.p.’ were inferred to be quantitatively less than any published number, given that ‘n.p.’ is primarily used to represent a number too small to publish. Thus, if an indicator was published as ‘n.p.’ in the first report and a numerical value in the second, the direction of change was taken to be an increase. Where both time points were published as ‘n.p.’, the indicator was inferred to have not changed. It was assumed that the threshold value for suppressing data was the same for both reports.

### 2.2. Scoring

#### 2.2.1. Unit Change Indicator (Did the Indicator Change over Time?)

The measurement of improvement was abstracted to a simple unit change indicator (*U*) reflecting the direction of change between time points *T1* and *T2* for each indicator, *i*:*U* = +1: The value of the indicator increased over the specified time period;*U* = 0: The value of the indicator did not change over the specified time period;*U* = −1: The value of the indicator decreased over the specified time period.
$$U\left({∆}_{i}\right)=\left\{\begin{array}{cc} -1 & {∆}_{i}<0\\ 0 & {∆}_{i}=0\\ 1 & {∆}_{i}>0 \end{array}\right.\;,$$
where ${∆}_{i}={T2}_{i}-{T1}_{i}$ for each indicator *i*.

#### 2.2.2. Confidence Indicators (What Is the Strength of Evidence for Any Observed Change?)

In this repeated measures framing, the percentage change over time presents a simple and transparent measure of relative change. This was simplified further to whether the percentage change met an arbitrary threshold, *τ*, presented as a binary indicator (*C*1).
*C*1 = 1: The percentage change of the indicator met or exceeded an arbitrary threshold;*C*1 = 0: The percentage change of the indicator was less than this threshold.
$$C1\left({\%∆}_{i},\;\tau \right)=\left\{\begin{array}{cc} 1 & {\%∆}_{i}\;\geq \;\tau \\ 0 & {\%∆}_{i}<\tau \end{array},\;\right.$$where $\%{∆}_{i}=100\left(\frac{{T2}_{i}-{T1}_{i}}{{T1}_{i}}\right)$, and τ is an arbitrary threshold.

In the absence of information on how much an indicator can be reasonably expected to change, the largest absolute difference (LAD) observed by any jurisdiction was taken as the estimator for the magnitude of change possible. An arbitrary proportion, *q*, of the LAD defines the threshold for a binary indicator of comparative change (*C*2).
*C*2 = 1: The magnitude of change of the indicator met or exceeded an arbitrary threshold defined by a stated proportion of the largest absolute change in the available data;*C*2 = 0: The percentage change of the indicator was less than this threshold.
$$C2\left({∆}_{i},\;q\right)=\left\{\begin{array}{cc} 1 & {\%∆}_{i}\;\geq \;\frac{\max(\left|{∆}_{i}\right|)}{q}\\ 0 & {\%∆}_{i}<\frac{\max(\left|{∆}_{i}\right|)}{q} \end{array}\right.\;,\;$$where *q* is an arbitrary threshold.

For this study, thresholds of *τ* = 2% relative change and *q* = 10% of LAD were selected for the confidence indicators. These values were arbitrary but expressed a preference for high sensitivity to changes.

#### 2.2.3. Improvement Score

Unit change and confidence indicators were combined with the expected direction of effect via summation to create a total improvement score between −3 and +3 for each indicator. The sign of the improvement score indicates the direction of effect (positive for improvement, negative for worsening), and the magnitude indicates confidence for the change.

### 2.3. Aggregation and Dashboard

Improvement scores were aggregated to HPF tiers and groups and averaged by the number of viable indicators available in the data for each state/territory. Standard errors for the mean improvement score were calculated. Improvement scores were visualised as bar charts, with average improvement displayed as green bars and average worsening as red bars. Charts were generated for each HPF tier within states/territories, with bars representing HPF groups. Error bars represented the 95% Confidence Interval, calculated as ±1.96 times the standard error of the mean improvement score. This is presented only to highlight variability in the underlying scores and not as a test of statistical significance.

### 2.4. Ethics

Although this study involved de-identified data obtained from free open-access websites, ethics approval was obtained from the Human Research Ethics Committee at Central Queensland University (Application Number 22739).

## 3. Results

An example of the calculation of change and confidence indicators is given in Table 2 for the indicator of self-assessed health status. This indicator was available across all jurisdictions for both timepoints and scored as a weighted average of the ordinal response categories (excellent = 5, very good = 4, good = 3, fair = 2, poor = 1). The percentage change threshold (*τ* = 2%) was exceeded for all observations and assigned ±1 for the first confidence indicator, except for Western Australia (WA). Total improvement scores ranged from 2 to −3 across this indicator. For this indicator, the direction of improvement was positive, and results are interpreted as improvement if the score is positive. According to these data, all jurisdictions observed improvement in self-reported health status except for SA and ACT. The results indicate less confidence in the WA results compared with other jurisdictions.

Table 3 shows the distribution of improvement scores against each change and confidence indicator configuration for all valid observations at the state/territory level (*n* = 2129). The majority (70.0%) of indicators qualified for both confidence indicators, reflecting the intentionally sensitive threshold settings of 2% relative change and one-tenth largest absolute difference. Improvement scores with an absolute value of 2 had comparable proportions of either the percentage change or LAD indicators (8.5% and 9.2%, respectively).

Improvement scores were aggregated by HPF groups and tiers using score means denominated by the number of indicators in the group. Table 4 demonstrates this calculation using the Tier 1 group ‘Life expectancy and wellbeing’ for Queensland. Three out of five indicators improved, resulting in an average improvement score of 0.8.

Appendix A provides an example of a whole-of-framework dashboard partitioned by state/territory and HPF tier. Average improvement scores and error intervals are overlaid for reference.

## 4. Discussion

This paper explores bridging the HPF as a theoretical construct and the publicly available data provided against its measures. A whole-of-framework, whole-of-system monitoring perspective was taken to summarise 289 eligible indicators at the state/territory level, organised by the HPF’s tier and group hierarchy. Data accompanying the 2017 and 2020 reports were used to compute improvement over time. Unit change and confidence indicators were developed to create an abstract but interpretable improvement score that was suitable for aggregation and visualisation at scale.

### 4.1. Unit Change Indicator

Improvement scores formed the basis of the method, simplifying hundreds of data tables into a single analytical framework based on whether indicators were, on average, improving or not. It is not argued that this method does not ignore other important information (e.g., effect size and contextual factors), only that the direction of change is the single most important piece of information to which every HPF indicator can be reduced; any question of by what magnitude a measure has improved implicitly asks in the first instance whether the measure has changed at all. Moreover, given the nature of low-denominator rates and high standard errors, measures of effect size, such as rate ratios or rate differences based on the HPF data, could generate misleading results, overemphasise outliers, and invite erroneous interpretations. The influence of outliers on unit change indicators and concepts like clinical significance and floor/ceiling effects are beyond the scope of this paper but warrant attention to further develop this approach.

### 4.2. Confidence Indicators

Confidence indicators (percentage change and largest absolute difference) were devised to be used similarly to statistical significance for reporting “meaningful” changes. When combined with the unit change indicator into a final improvement score, directional information remains key, while the score’s magnitude represents confidence in that directional information; a score of 3 is greater in confidence than a score of 1, and a score of −3 is equal in confidence to a score of +3. Admittedly, both confidence indicators were based on the magnitude of changes—the change is given confidence only because it is large enough—unlike statistical confidence measures, which are mostly denominated by variance with associated probability distributions and where the change is given confidence mostly because the sample is large enough.

Small values and zero counts at the first time point are problematic for the percentage change indicator (C1), leading to extreme values or divisions by zero. For example, the proportion of government-funded vocational qualifications completed in the Australian Capital Territory (ACT) increased from 0.02% to 1.40% across reports generating a 6690% relative improvement. This highlights the benefit of the method in constraining outliers and extreme values and the confirmatory approach of using both relative (percentage change) and comparative (LAD) indicators to assess confidence. The ACT also displayed the largest absolute difference of 1.38% for this indicator, and so was awarded both confidence indicators and a total improvement score of +3.

Divisions by zero were not addressed by the current method. There were 21 indicators that reported zero values in the 2017 report and 25 in the 2020 report. Implementations in different programming languages will need to consider how the specific language handles the problem. Python implementations return “inf” (for infinity) values, which were converted to empty values (“NaN” for “not a number”) for the percentage change confidence indicator. These observations relied solely on the LAD confidence indicator for appraisal, limiting the possible improvement score to ±2. Alternatively, leaving the result as an infinity value would ensure that the percentage change indicator was always awarded highlighting fringe cases for which interpretation is not straightforward, even at the level of abstraction pursued by this method.

An artefact of the LAD indicator (C2) is that it will always be awarded to the jurisdiction reporting the largest difference. In practice this difference may be trivial, inferring no jurisdiction was observed to improve or worsen in practical terms on that indicator but for which all were awarded C2. Trivial LADs may set a ‘low bar’ in qualifying for the C2 indicator.

Where one or both timepoints were published as ‘n.p.’, the computation of any confidence indicators was precluded. However this is compatible with the motivation of the confidence indicators—given unpublished values, there cannot be confidence in the imputed difference. Differences would typically be analysed using statistical tests; however, due to the breadth and diversity of sources of data, different measurement scales and formatting decisions regarding value suppression and missing data in the published data tables, it was not immediately apparent which tests are appropriate and for which measures, even for simple pairwise comparisons. Confidence indicators could be replaced by statistical significance (i.e., a binary interpretation of meeting the threshold for significance) if, upon appropriate scrutiny, a proper statistical test can be defensibly used (e.g., assumptions satisfied) for an indicator’s change over time. For example, if a regression model is fitted for time-series data, the statistical significance of the time coefficient can serve as the basis for the confidence indicator.

### 4.3. Aggregation

Systematic effects cannot be adequately appraised without multiple observations. In the HPF data, the number of observations per indicator is limited by the number of jurisdictions, availability of data, and publication choices. In this analysis, the theoretical structure of the HPF was leveraged, aggregating improvement scores into HPF groups and tiers. Empirically validating (e.g., via principal components or other dimension-reduction methods) the tier and group structure of the HPF, and/or the AIHW’s allocation of data against the HPF structure, is beyond the scope of this paper and likely impossible with publicly available data. The allocation of data as given by the AIHW was assumed to be appropriate for this exercise, except where indicators were duplicated across measures and tiers. For example, hospitalisations for mental-health-related conditions being allocated to both Tier 1 (as a health outcome) and Tier 3 (as an indication of system performance), whereas they were allocated to Tier 1 for this study in line with other hospitalisation data.

Aggregating to group level provides between 2 and 12 measures (except for one group “Person-related factor”, which has only one measure—2.22 Overweight and obesity), and aggregating to tier level provides 22–24 measures (see Table 1). Each measure can also have multiple improvement scores, up to as many as there are published data tables, in theory. This presents the possibility of improvement score distributions that may have potentially meaningful statistical properties. This is a topic for additional research.

The simplest aggregating methods of summation and averaging were employed in the current approach; however, other options are likely possible. In this study, unit change indicators were used to represent the direction of change in HPF indicators, independently of effect size. The unit vector in linear algebra is used to represent direction independently of magnitude. This suggests different aggregation or summarising methods for unit change indicators can be explored using methods such as vector magnitude or similarity/distance measures. The appropriateness and comparison of such methods are beyond the scope of this paper and present interesting questions for future work.

### 4.4. Observation and Reporting Periods

The timing of observations across indicators was not addressed in this method. The data sources from which HPF tables are developed operate on different collection/reporting schedules, ranging from standalone reports, discontinued collections, and periodic samples to ongoing daily registrations, and different formatting decisions are made for publication. For example, HPF data provide aggregated 3-year and 5-year rates for hospitalisations and deaths, respectively, in the consecutive 2017 and 2020 reports. For mortality, these observation periods partially overlap, creating dependence between observations (some deaths are counted in both periods).

The method presented in this manuscript used relative improvement for each indicator in a repeated measures format, but aggregating improvement scores by group and tier may facilitate the perception of standardised reporting periods that invite inferences about the relationship between groups and tiers that would not hold under rigorous analysis. Little can be done to accommodate this analytically, at least not without the cost of greatly complicating the analysis and interpretation, thereby contradicting the original motivation for this approach.

### 4.5. Geography

The state/territory level allowed for the maximum number of observations across the HPF; however, some data sources, notably the National Key Performance Indicators (nKPI), aggregated smaller jurisdictions into neighbouring states. Victoria and Tasmania are reported as a single jurisdiction, as are the ACT and New South Wales (NSW). One measure in HPF (Illicit drug or substance use during pregnancy) reported Tasmania and the ACT as a single aggregate. Consequently, these indicators could not be appropriately allocated when partitioning by state/territory and were not included in the dashboard’s group averages. Similarly, indicators for which only national estimates were available could not be allocated. In all cases, however, improvement scores were calculable, and whether to average over states, remoteness, or national figures in a dashboard or other summary is a design consideration.

### 4.6. Summary

Overall, this analytical method is considered exploratory in nature and similar to that of an omnibus test—intended to highlight important global features upon which to design more detailed studies. HPF data are collated by the AIHW from multiple repositories. This use of secondary data inevitably invites acknowledgments of what an analysis cannot say, owing to methods of collection, sampling bias, or unobserved variables, and the standard mantra regarding correlation not being causation (though no attempt has been made here to infer relationships between indicators, groups, or tiers).

## 5. Conclusions

For the potential of the HPF to be fully realised, a method of synthesising data from multiple sources that use different levels of measurement into a single high-level analytical framework is required. This would extend the utility of the HPF data beyond an expansive directory of individual indicators into a whole-of-system, whole-of-framework monitoring capability. Such a method would ideally produce results that are easily communicated and understood.

## Figures and Tables

**Table 1 ijerph-21-00340-t001:** Health Performance Framework hierarchical structure.

Tier	Group	Measures
Tier 1—Health status and outcomes	Health conditions	1.01 Low birth weight, 1.02 Top reasons for hospitalisation, 1.03 Injury and poisoning, 1.04 Respiratory disease, 1.05 Circulatory disease, 1.06 Acute rheumatic fever and rheumatic heart disease, 1.07 High blood pressure, 1.08 Cancer, 1.09 Diabetes, 1.10 Kidney disease, 1.11 Oral health, 1.12 HIV/AIDS, hepatitis, and sexually transmissible infections
	Human function	1.13 Community functioning, 1.14 Disability, 1.15 Ear health, 1.16 Eye health
	Life expectancy and wellbeing	1.17 Perceived health status, 1.18 Social and emotional wellbeing, 1.19 Life expectancy at birth
	Deaths	1.20 Infant and child mortality, 1.21 Perinatal mortality, 1.22 All-causes age-standardised death rates, 1.23 Leading causes of mortality, 1.24 Avoidable and preventable deaths
Tier 2—Determinants of health	Environmental factors	2.01 Housing, 2.02 Access to functional housing with utilities, 2.03 Environmental tobacco smoke
	Socio-economic factors	2.04 Literacy and numeracy, 2.05 Education outcomes for young people, 2.06 Educational participation and attainment of adults, 2.07 Employment, 2.08 Income, 2.09 Index of disadvantage
	Community capacities	2.10 Community safety, 2.11 Contact with the criminal justice system, 2.12 Child protection, 2.13 Transport, 2.14 Indigenous people with access to their traditional lands
	Health behaviours	2.15 Tobacco use, 2.16 Risky alcohol consumption, 2.17 Drug and other substance use including inhalants, 2.18 Physical activity, 2.19 Dietary behaviour, 2.20 Breastfeeding practices, 2.21 Health behaviours during pregnancy
	Person-related factor	2.22 Overweight and obesity
Tier 3—Health system performance	Effective, appropriate, efficient	3.01 Antenatal care, 3.02 Immunisation, 3.03 Health promotion, 3.04 Early detection and early treatment, 3.05 Chronic disease management, 3.06 Access to hospital procedures, 3.07 Selected potentially preventable hospital admissions, 3.08 Cultural competency
	Responsive	3.09 Discharge against medical advice, 3.10 Access to mental-health services, 3.11 Access to alcohol and drug services, 3.12 Aboriginal and Torres Strait Islander people in the health workforce, 3.13 Competent governance
	Accessible	3.14 Access to services compared with need, 3.15 Access to prescription medicines, 3.16 Access to after-hours primary health care
	Continuous	3.17 Regular general practitioner or health service, 3.18 Care planning for chronic diseases
	Capable	3.19 Accreditation, 3.20 Aboriginal and Torres Strait Islander people training for health-related disciplines
	Sustainable	3.21 Expenditure on Aboriginal and Torres Strait Islander health compared with need, 3.22 Recruitment and retention of staff

Source: Australian Health Ministers’ Advisory Council, 2017, Aboriginal and Torres Strait Islander Health Performance Framework 2017 Report, AHMAC, Canberra.

**Table 2 ijerph-21-00340-t002:** Example of improvement score calculation with self-assessed health status.

							Confidence Indicators		
State	Time 1	Time 2	Δ	LAD	% Change	Unit Change	C1 (% Change)	C2 (LAD)	Improvement Score
ACT	63.7	60.7	−3.0	3.9	−4.6	−1	−1	−1	−3
NSW	63.4	65.7	2.3	3.9	3.6	1	1	1	3
NT	67.1	69.1	2.0	3.9	2.9	1	1	1	3
Qld	63.7	65.8	2.1	3.9	3.3	1	1	1	3
SA	63.3	60.7	−2.6	3.9	−4.2	−1	−1	−1	−3
Tas	61.4	63.3	1.9	3.9	3.1	1	1	1	3
Vic	62.7	66.6	3.9	3.9	6.2	1	1	1	3
WA	63.9	65.1	1.2	3.9	1.9	1	0	1	2

**Table 3 ijerph-21-00340-t003:** Distribution of improvement scores by unit change and confidence indicators—absolute values.

Indicators			Improvement Score				
Unit Change	% Change	LAD	0	1	2	3	Total
0	0	0	77 (3.6%)				
1	0	0		183 (8.6%)			
1	0	1			196 (9.2%)		
1	1	0			182 (8.5%)		
1	1	1				1491 (70.0%)	
Total			77 (3.6%)	183 (8.6%)	378 (17.8%)	1491 (70.0%)	2129

**Table 4 ijerph-21-00340-t004:** Group aggregation example using life expectancy and wellbeing—Queensland.

Indicator	Improvement Score	Group Mean	SEM
Self-assessed health weighted rating	3	0.8	1.4
Low/moderate distress level	−2		
Life expectancy—Females	3		
Life expectancy—Males	3		
Mental-health related hospitalisations	−3		

## Data Availability

All data used in the manuscript are publicly available on the dedicated website indigenoushpf.gov.au in the form of summary reports and accompanying data tables (Microsoft Excel workbooks).

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
