# Peer review of "The Utility of Data Collected as Part of Australia’s Aboriginal and Torres Strait Islander Health Performance Framework"

_ijerph, 2024, doi:10.3390/ijerph21030340_

Round 1
Reviewer 1 Report
Comments and Suggestions for Authors
I found this paper to be interesting, well written, and addressing an important gap in the literature. The authors proposed a methodology that could be used to better track health and system performance using publicly available and regularly reported data that are otherwise difficult to synthesise. The use of this or comparable methods has the potential to better inform health and other services’ policy making. As the authors noted, this methodology is exploratory and further work is required to validate its utility and test for levels of arbitrary thresholds (which conceivably may differ for different indicators). Nevertheless, I agree that this type of approach could be useful for informing policy, research etc.
The example provided was instructive and was explained in a way that was reasonably straightforward to follow.
My suggestions for improving the paper are relatively minor.
For example, in the Methodology section the authors write:
“Unit change and confidence indicators were combined with the expected direction of effect to create a total improvement score between -3 and +3 for each indicator.”
It was not stated how they were combined. Studying the example allowed me to determine that these three measures were summed. This could be stated more clearly.
For the text related to Table 4, the authors state:
Three out of five indicators improved resulting in an average improvement score of 0.8.
Perhaps an explanation for how the average improvement score was attained might be helpful though it wasn’t difficult to work out as being =∑improvement scores in a group/number of indicators in a group.
Example Dashboard (Figure 1) has a little incongruity with Table 1. Eg. Table 1 doesn’t include Accessibility as a Health System Performance group, but it is included in Figure 1. Table 1 doesn’t include Socio-economic factors as a Determinants of Health group, but it is included in Figure 1.
Slight issue with formatting 2.3 Aggregation and dashboard section. Line 156. The word Dashboard appears as a stand alone sentence. Was this meant to be a sub-heading?
Comments on the Quality of English LanguageSee above.
Author Response
Provided in attached word document

Reviewer 2 Report
Comments and Suggestions for Authors
My comments for revision/edits are relativeely minor:
- the word "improve" is used frequently and perhaps should be change, thus not indicating the direction. As in all cases, improve might not be the appropriate word
- Table 2 - not clear why Tier 3 improvement score are not provided
- Table 3 - not clear why column is identified as % change, yet absolute value in indicated in the top heading
- Final paragraph provides limitation to secondary data analysis and references method of data collection. The methods of collection to create this data set should be explained with a bit more detail. Was it derived from data tables from 2017 and 2020 HPF reports? Not clear what entity is developing HPF the reports.
Author Response
Please refer to attached word document
